# Implementation of colorectal cancer screening interventions in low-income and middle-income countries: a scoping review protocol

Désirée Schliemann  ,[1] Nicholas Matovu,[1] Kogila Ramanathan,[2] Paloma Muñoz-Aguirre,[3] Ciaran O'Neill,[1] Frank Kee,[1] Tin Tin Su,[2] Michael Donnelly[1]

[1]Centre for Public Health and UKCRC Centre of Excellence for Public Health, Queen's University Belfast, Belfast, UK
[2]South East Asia Community Observatory (SEACO), Jeffrey Cheah School of Medicine and Health Sciences, Monash University Malaysia, Subang Jaya, Malaysia
[3]Centre for Research and Population Health, Instituto Nacional de Salud Publica, Cuernavaca, Morelos, Mexico

**Correspondence to**
Dr Désirée Schliemann;
D.Schliemann@qub.ac.uk

## ABSTRACT

**Introduction** Colorectal cancer (CRC) imposes a significant global burden of disease. CRC survival rates are much lower in low-income and middle-income countries (LMICs). Screening tends to lead to an improvement in cancer detection and the uptake of available treatments and, in turn, to better chances of cancer survival. Most evidence on CRC screening interventions comes from high-income countries. The objective of this scoping review is to map the available literature on the implementation of CRC screening interventions in LMICs.

**Methods and analysis** We will conduct a scoping review according to the framework proposed by Arksey and O'Malley (2005). We will search MEDLINE, EMBASE, Web of Science and Google Scholar using a combination of terms such as "colorectal cancer", "screening" and "low-middle-income countries". Studies of CRC screening interventions/programmes conducted in the general adult population in LMICs as well as policy reviews (of interventions in LMICs) and commentaries on challenges and opportunities of delivering CRC screening in LMICs, published in the English language before February 2020 will be included in this review. The title and abstract screen will be conducted by one reviewer and two reviewers will screen full-texts and extract data from included papers, independently, into a data charting template that will include criteria from an adapted template for intervention description and replication checklist and implementation considerations. The presentation of the scoping review will be reported according to the Preferred Reporting Items for Systematic Reviews and Meta-Analysis extension for Scoping Reviews guidance.

**Ethics and dissemination** There are no ethical concerns. The results will be used to inform colorectal screening interventions in LMICs. We will publish the findings in a peer-reviewed journal and present them at relevant conferences.

## Strengths and limitations of this study

► This scoping review will map the literature on colorectal cancer screening in low-income and middle-income countries, thus, highlighting opportunities and challenges for those countries and informing future global health research.
► The approach presented here will provide a timely scoping review synthesis of the literature in this field.
► The inclusion criteria are broad in order to 'paint' a comprehensive picture.
► We will include only studies published in the English language, which may mean that programmes and interventions published in other languages will be missed.
► The review uses the World Bank classification rather than the Human Development Index (HDI) to define middle-income countries. Some countries that the HDI define as high-income countries may be classified as upper-middle income by the World Bank.

## INTRODUCTION

Colorectal cancer (CRC) is the fourth most common cancer worldwide with an age-standardised incidence rate of 19.7/100 000 for men and women combined and it remains the third most common cause of cancer deaths globally with an estimated age-standardised mortality rate of 8.9% in 2018.[1] Large disparities exist in CRC incidence and mortality rates between high-income and low-income countries. In Oceania, Europe and North America, CRC mortality rates are about one-quarter of incidence rates, whereas in Asia, Latin America and the Caribbean mortality rates represent over half of incidence rates, and in Africa, mortality rates are over two-thirds of incidence rates.[1] CRC incidence and mortality correlate with the Human Development Index (HDI). Low-income and middle-income countries (LMICs) have experienced a rapid increase in CRC incidence and mortality rates, whereas in high-income countries the rates are stabilising or decreasing.[2] Rapid yearly increases in CRC incidence of up to 2% have been recorded for some middle-income countries such as Brazil and Costa Rica since the early 2000s[3] and in mortality rates in the Philippines (5.7%) and Belarus

(3.7%) over a 10-year period.[2] It has also been estimated that CRC incidence in Africa is expected to increase by 85% by 2030.[4] Arguably, ageing populations and lifestyle changes that more commonly resemble Western lifestyles in LMICs are the main reasons for the increasing trends as well as advancements in cancer detection.[2]

The International Agency for Cancer Research suggests that there is sufficient evidence in relation to reduced CRC mortality to recommend screening (biannually) for CRC.[5] Screening is most effective when it is delivered in a way that detects cancer at an early stage, that is stage 1 or 2,[6] when CRC patients are often asymptomatic. The guaiac faecal occult blood test (gFOBT) and the faecal immunochemical test (FIT) for haemoglobin are non-invasive, inexpensive stool tests that screen for small amounts of blood and that tend to be offered to asymptomatic, 'at-risk' individuals (eg, people at a certain age or with a CRC family history). Invasive, visual screening techniques such as colonoscopy and flexible sigmoidoscopy are offered, generally, to symptomatic and high-risk individuals or those with a positive gFOBT/FIT results. In the last 10 years, many high-income countries have introduced organised, population-based cancer screening programmes and follow a systematic approach whereby anyone over a certain age (eg, 50 years) is invited to complete a stool test at regular (eg, 2 yearly) intervals. Screening interventions are recommended only if high-quality treatment and follow-up can be delivered[5] and, therefore, the readiness of a country's health system to implement a screening programme is crucial.[7] Some middle-income countries offer opportunistic screening (eg, sporadic stool tests to individuals at-risk who attend a clinic) but lack financial resources and the infrastructure to support organised screening (ie, qualified staff, screening centres and cancer registries). Low-income countries appear to have no or limited CRC screening or treatment facilities in place.[8] Many LMICs also lack cancer registries and, therefore, recording of cases and cancer control efforts are a major challenge.[9–11] Thus, evaluation reports of CRC screening interventions come mainly from high-income countries and most CRC interventions and programmes were delivered at least partly through the post, that is invitation letters to attend screening and/or mailed stool-test kits to be completed at home.[12 13] The discrepancies between CRC incidence globally and the availability of opportunistic or population-based screening remain a concern[8] and the rising CRC incidence and high mortality rates in LMICs countries require efforts to improve CRC outcomes.

### Review objectives
The purpose of this scoping review is to review the global health literature, map evidence from peer-reviewed and grey literature, examine programmes and interventions that aim to improve CRC screening uptake in LMICs and highlight research gaps and opportunities in order to answer the three review questions below. Findings will be used to guide future CRC screening efforts in countries

with limited resources. We will use the classification by the World Bank to define countries in terms of lower-middle income, upper-middle income and low-income economies (online supplementary materials 1).

## METHODS AND ANALYSIS
We will follow the research approach that conforms with the Preferred Reporting Items for Systematic Reviews and Meta-Analysis extension for Scoping Reviews.[14] A scoping review is the most appropriate type of review, given the underexplored nature of this area and the need for iterative-style searching of multiple sources. The scoping review will be guided by the methodological five-step framework developed by Arksey and O'Malley[15] as outlined below.

### Stage 1: identifying the research question
The review will aim to scope literature in the public domain regarding (1) the content delivered, implementation and uptake of CRC screening interventions and programmes in LMICs and (2) challenges and opportunities for CRC screening in LMICs. In particular, the review will attempt to answer the following research questions:
1. What interventions/programmes have been implemented in LMICs that were designed explicitly to encourage people to attend CRC screening (including interventions implemented at a patient-level, provider-level and system-level)?
2. What are the opportunities and challenges in terms of implementing CRC screening interventions/programmes in LMICs?
3. What are the findings (qualitative and quantitative outcomes) from interventions/programmes in terms of, for example, implementation, reach, uptake, engagement (including differences between rural and urban areas) and costs/resources?

A scoping rather than a systematic review was the chosen methodological approach as this topic is unexplored and, so, the main aim of the review is to scope the landscape of relevant reports and to address the above-noted broad questions as well as identify whether or not there is a need, or there are sufficient studies, to conduct a systematic review of the effectiveness or cost-effectiveness of CRC screening interventions.

### Stage 2: identifying relevant studies
We will search MEDLINE, EMBASE, Web of Science, and Google Scholar for relevant literature. Furthermore, we will hand search the reference lists of relevant reviews and studies and Google Web for unpublished reports and briefings. In addition, we will contact experts in the field of CRC screening in LMICs to identify additional programmes that may not have been published in the scientific literature. We have devised an initial search strategy (with the expertise of an experienced subject librarian) for MEDLINE (online supplementary material 2) which will be adapted for other databases. Search terms

revolve around the three categories related to "colorectal cancer", "screening" and "LMICs".

## Stage 3: study selection

The lead author (DS) will conduct the title and abstract screen and two authors (DS and NM or PM-A) will separately screen the retrieved full-texts for inclusion in this scoping review based on broad inclusion and exclusion criteria. Inconsistencies between reviewers will be discussed with a third reviewer (MD).

This scoping review will focus on two types of literature: (1) dedicated empirical studies of CRC screening interventions or programmes that were explicitly designed to encourage the public to attend CRC screening and (2) commentaries/editorials and policy reviews that discuss opportunities and challenges for CRC screening implementation in LMICs. National screening programmes as well as screening programmes and interventions delivered at lower population aggregates and programmes of any duration published before February 2020 will be included. Only studies conducted in LMICs targeting the general adult population (18 years or older) or asymptomatic populations 'at-risk' for CRC will be considered for inclusion (this may vary between countries, but is often referred to as adults aged over 40 or 50 years). Studies aiming to improve cancer screening among cancer patients and healthcare professionals will be excluded from this review, given their advanced understanding and direct experience of the importance of cancer screening compared with the general population. However, interventions targeted at healthcare professionals with the aim of improving screening among the general population will be considered for inclusion. We will exclude protocols if no further information on the implementation of the programme is available. We will only include studies published in the English language and will not apply restrictions concerning the year of publication.

## Stage 4: charting the data

We will chart the data from the included intervention studies/programme in a descriptive manner according to categories in an adapted Template for Intervention Description and Replication (TIDieR) checklist for the reporting of interventions[16] in order to illuminate key components and facilitate the production of standardised descriptions. The following fields will be added to the domains of the TIDieR checklist and charted accordingly: 'reference information', 'study design' and 'for whom' (ie, target population) (table 1). We will also chart data descriptively on implementation considerations to identify key concepts that influence dissemination and successful implementation of evidence-based programmes. This will be guided by a template developed by Tierney et al[17] which includes 10 factors (table 2), one of which overlaps

**Table 1** Data charting domains and explanations adapted from the Template for Intervention Description and Replication checklist

| Data charting domain | Explanation* |
|---|---|
| Reference information | Extract first author and year of publication |
| Brief name | Name or a phrase that describes the intervention |
| Why | Describe any rationale, theory or goal of the elements essential to the intervention |
| What | Study design: for example, cross-sectional, quasi-experimental and observational |
| | Materials: describe any physical or informational materials used in the intervention, including those provided to participants or used in intervention delivery or in training of intervention providers. Provide information on where the materials can be accessed (such as online appendix, URL) |
| | Procedures: describe each of the procedures, activities and/or processes used in the intervention, including any enabling or support activities |
| Who provided | For each category of intervention provider (such as psychologist and nursing assistant), describe their expertise, background and any specific training given |
| How | Describe the modes of delivery (such as face to face or by some other mechanism, such as internet or telephone) of the intervention and whether it was provided individually or in a group |
| Where | Describe the type(s) of location(s) where the intervention occurred, including any necessary infrastructure or relevant features |
| For whom | Describe the target audience as well as inclusion and exclusion criteria if provided. Provide information about the sampling procedure. |
| When and how much | Describe the number of times the intervention was delivered and over what period of time including the number of sessions, their schedule and their duration, intensity or dose |
| Tailoring | If the intervention was planned to be personalised, titrated or adapted, then describe what, why, when and how |
| Modifications | If the intervention was modified during the course of the study, describe the changes (what, why, when and how) |

*Data may not be provided in each study. All information provided that is listed here will be extracted descriptively.

**Table 2** Data charting domains and explanations from implementation science frameworks

| Data charting domain* | Explanation |
|---|---|
| Acceptability | Perception among implementation stakeholders that the intervention is agreeable, palatable or satisfactory |
| Adoption | Intention, initial decision or action to try or employ an innovation or evidence-based practice |
| Appropriateness | Perceived fit, relevance or compatibility of the intervention for a given practice setting, provider, target population or problem |
| Feasibility | Extent to which the intervention can be successfully used or carried out within a given setting |
| Fidelity | Degree to which an intervention was implemented as it was prescribed in the original protocol or as it was intended (planned and actual fidelity) |
| Implementation cost | Cost impact of an implementation effort (influenced by intervention complexity, implementation strategy and setting) |
| Intervention complexity | Perceived difficulty of implementation, reflected by duration, scope, radicalness, disruptiveness, centrality and intricacy and number of steps required to implement |
| Penetration | Integration of a practice within a service setting and its subsystems |
| Reach | The absolute number, proportion and representativeness of individuals who are willing to participate in an intervention |
| Sustainability | Extent to which a newly implemented intervention is maintained or institutionalised within a service setting's ongoing, stable operations |

*Data may not be provided in each study. All information provided that is listed here will be extracted descriptively.

with the TIDieR checklist (ie, intervention 'fidelity') and which therefore will only be addressed under implementation considerations. Both checklists have been piloted and deemed appropriate by the team. Two authors (DS and NM, KR or PM-A) will independently extract information and any discrepancies will be resolved in discussion with a third author (MD or TTS). In addition, themes will be extracted qualitatively from reviews and commentaries/editorials about the implementation challenges and opportunities of CRC screening programmes in LMICs.

### Stage 5: collating, summarising and reporting the results

We will summarise the data extracted from each study and present the findings in a series of detailed tables that will be organised according to the TIDieR and implementation science domains, and marshalled to the content of each review question (eg, intervention) and then grouped or aggregated according to country, region and income level. In particular, the review will present the barriers and enablers to the successful implementation and uptake of CRC screening in LMICs and resource-constrained settings.

### PATIENT AND PUBLIC INVOLVEMENT

The results of the protocol will be shared and discussed with key stakeholders in various global health studies, particularly in relation to our studies in rural communities in Malaysia (ie, the Ministry of Health and non-governmental cancer organisations that provide CRC screening services).

### ETHICS AND DISSEMINATION

This paper presents the protocol for a scoping review of CRC screening interventions in LMICs. Ethics approval is not necessary as the data will be collected from publically available resources. This review will advance the knowledge on CRC screening programmes for researchers, general practitioners, public health practitioners and policy makers working in LMICs. The results will be disseminated through a peer-reviewed journal and presented at relevant conferences. Furthermore, this scoping review will be conducted to inform the implementation of a CRC pilot screening intervention for Malaysia.

**Contributors** DS, MD and TTS conceived the idea for the scoping review. DS and MD developed the research question. DS, NM, KR, PM-A, CO, FK, TTS and MD aided in developing the scoping review methods. DS drafted the manuscript and MD led the editing of the manuscript. All authors revised the manuscript critically for intellectual content and gave final approval of the submitted and revised manuscript.

**Funding** This research is funded by the Medical Research Council (UK) Global Challenges Research Fund.

**Competing interests** None declared.

**Patient consent for publication** Not required.

**Provenance and peer review** Not commissioned; externally peer reviewed.

**ORCID iD**
Désirée Schliemann http://orcid.org/0000-0002-8746-3002

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
