## [Reviewer comments · BMJ Open]

ARTICLE DETAILS

TITLE (PROVISIONAL)	Implementation of colorectal cancer screening interventions in low- and middle-income countries: a scoping review protocol
AUTHORS	Schliemann, Desiree; Matovu, Nicholas; Ramanathan, Kogila; Muñoz, Paloma; O'Neill, Ciaran; Kee, Frank; Su, Tin; Donnelly, Michael

VERSION 1 – REVIEW

REVIEWER	Kevin Selby University of Lausanne, Switzerland
REVIEW RETURNED	02-Apr-2020

GENERAL COMMENTS	Thank you for this timely systematic review protocol. As the authors correctly state, there is an increasing burden of CRC in LMICs and an overview of existing evidence is needed. I agree that a scoping review is appropriate. This protocol needs some important improvements however: - Abstract: Study question unclear from abstract. You are focusing on any article related to CRC screening in LMICs? Or primarily interventional studies reporting results?- Intro line 10: 9.2% of what?- "Review objectives": I find it unclear whether you want to describe programmes and practices that are in place, or primarily include interventional studies (RCTs, before-after interventions, etc). It isn't clear to me what type of studies you expect to find, which is important afterwards for what type of analyses will be done. For instance, primarily at patient, provider or system level?"Stage 1": Here again it is unclear to me whether you are primarily targetting reports of programs, reports of screening outcomes, or reports of interventions to increase screening uptake. Even if you are unsure what you will find, it is still helpful to say what study types you are targetting.- Please specify that this is for screening, therefore the asymptomatic population.- What data synthesis is planned?- What quality appraisal is planned?- Any formal means of making recommendations based on collected evidence?- Please also address study limitations
---

REVIEWER	Elise Mansfield University of Newcastle, Australia
REVIEW RETURNED	16-Apr-2020

GENERAL COMMENTS	This is a well-written protocol exploring the literature on an important topic. I did have a few suggestions for improvement:
---

	I think it would be informative to have an additional review question looking at the effectiveness of described interventions in increasing CRC screening rates. This domain could be added to Table 1. The review inclusion/exclusion criteria needs to be more clearly described. It needs to be made clearer which types of studies will be included - only those which have evaluated interventions, or those which simply describe an intervention? I wasn't clear why studies focusing on patients or health care providers were excluded. There may be some types of interventions that involve a focus on patients attending specific health services or encouraging providers to recommend screening to their patients. Justification for the exclusion of these types of studies is needed. Additional clarification on the definition of populations 'at risk' of CRC should also be provided. Under Stage 4, it would be helpful to have more clarity on how the information about implementation considerations will be extracted from studies (e.g. are studies given a score for each criterion, or is presence/absence of information relating to the criterion coded?)
--	--

VERSION 1 – AUTHOR RESPONSE

Reviewer comments	Author reply	Page
First reviewer - Abstract: Study question unclear from abstract. You are focusing on any article related to CRC screening in LMICs? Or primarily interventional studies reporting results?	We have clarified the abstract.	Page 2
-Introduction, 9.2% of what?	We have corrected the statement.	Page 3
- "Review objectives": I find it unclear whether you want to describe programmes and practices that are in place, or primarily include interventional studies (RCTs, before-after interventions, etc). It isn't clear to me what type of studies you expect to find, which is important afterwards for what type of analyses will be done. For instance, primarily at patient, provider or system level?	We have included a more specific explanation to the protocol, in particular under ' stage 1 and study selection'.	Page 5-7
"Stage 1": Here again it is unclear to me whether you are primarily targetting reports of programs, reports of screening outcomes, or reports of interventions to increase screening uptake. Even if you are unsure what you will find, it is still helpful to say what study types you are targetting.	We further outlined this under stage 1 and stage 3.	Page 6 and 7
- Please specify that this is for screening, therefore the asymptomatic population.	We have included this under 'study selection'.	Page 7, stage 3

- What data synthesis is planned?	We highlighted what data we will synthesise under 'data charting'.	Page 7-8
- What quality appraisal is planned?	Since this is a scoping review, we will not conduct a quality appraisal at this stage. If we find sufficient studies that describe effectiveness/ cost-effectiveness, then we will conduct a systematic review or meta-analysis summarising that data and conduct a quality appraisal.	n/a
- Any formal means of making recommendations based on collected evidence?	We have added information, i.e. that the results will be used to inform a CRC screening intervention for Malaysia.	Page 9
- Please also address study limitations	The last two points under strengths and limitations refer to study limitations.	Page 3
Second reviewer I think it would be informative to have an additional review question looking at the effectiveness of described interventions in increasing CRC screening rates. This domain could be added to Table 1.	Thank you very much for your comments, similar to the point made above, since this is a scoping review, we will not examine effectiveness at this stage, at least not in the conventional or usual way that effectiveness is considered in systematic reviews. A systematic review or meta-analysis including a quality appraisal may be a second stage in our review work if it turns out that there are sufficient relevant studies focused on answering effectiveness and cost-effectiveness.	
The review inclusion/exclusion criteria needs to be more clearly described. It needs to be made clearer which types of studies will be included - only those which have evaluated interventions, or those which simply describe an intervention? I wasn't clear why studies focusing on patients or health care providers were excluded. There may be some types of interventions that involve a focus on patients attending specific health services or encouraging providers to recommend screening to their patients. Justification for the exclusion of these types of studies is needed. Additional clarification on the definition of populations 'at risk' of CRC should also be provided.	We have included further justification under 'study selection'.	Page 7
Under Stage 4, it would be helpful to have more clarity on how the information about implementation considerations will be extracted from studies (e.g. are studies given a score for each criterion, or is	Data will be charted in a descriptive manner. We have added this information to the protocol.	Page 8, 13, 14

presence/absence of information relating to the criterion coded?)		
---	--	--

VERSION 2 – REVIEW

REVIEWER	Kevin Selby University of Lausanne, Switzerland
REVIEW RETURNED	13-May-2020

GENERAL COMMENTS	Thank you for addressing my primary concerns. I still believe that a critical appraisal of the individual articles should be an important component of any systematic review, especially scoping reviews that consider multiple types of evidence. However, I do not think that this disagreement should be a barrier to publication. Likely, this step will be requested prior to publication. It would be better to specify your criteria for evaluating papers in your protocol.
--